# Conceptualizing Sleep Satisfaction: A Rapid Review

**DOI:** 10.3390/bs14100942

**Published:** 2024-10-14

**Authors:** Cleo Protogerou, Valerie Frances Gladwell, Colin R. Martin

**Affiliations:** 1Department of Psychology, University of Crete, 74150 Rethymno, Greece; 2Institute of Health and Wellbeing, University of Suffolk, Ipswich IP4 1QJ, UK; v.gladwell@uos.ac.uk (V.F.G.); c.martin6@uos.ac.uk (C.R.M.)

**Keywords:** sleep satisfaction, sleep quality, construct definition, rapid review, systematic process

## Abstract

Good, satisfying, sleep is a key indicator and determinant of health and wellness. However, there is no consensus about how to define and measure good sleep. The present research aimed to define sleep satisfaction through the extant literature and disentangle it from sleep quality, a conceptually similar construct. Systematic review methods were adapted for a rapid review approach. The entire review was completed in eight weeks. Tabulation coding with content analysis was used to identify key categories and synthesize findings. A systematic process for generating construct definitions was followed. Database search yielded 51 eligible studies (*N* > 218,788), representing diverse adult populations, in 20 countries. Designs varied in rigour. Sleep satisfaction was defined as a personal, introspective, and global judgment about one’s feelings of contentment with one’s sleep, at a particular point in time. Sleep satisfaction was understood as an indicator of general health, impacted by and varied as a function of one’s sleep environment and individual-level characteristics. This rapid review contributes to the literature by providing the first systematically generated definition of sleep satisfaction, with strong implications for measurement, research, and practice.

## 1. Introduction

Good sleep plays a critical role in health and wellness, with the bulk of extant evidence demonstrating its protective role from virtually all disease states, accidents, and all-cause mortality [1,2]. However, globally, sleep disturbances are reported by 15 to 50 % of adults [1], and sleep disorders are a frequently encountered clinical condition [3]. 

The research investigating the linkages between sleep and health has, overwhelmingly, focused on the prevalence, aetiology and improvement of disturbed and disordered sleep [4,5]. Related, several measures exist to assess people’s reports and physiology of disturbed sleep; collectively these are known as *sleep quality* measures (for reviews of such measures see Fabbri et al., 2021 [6] and Ibáñez et al., 2018 [7]). Several sleep disorder-specific measures also exist (e.g., Bastien et al., 2001 [8]; Dietmann et al., 2021 [9]; Johns, 1994 [10]; Lichstein et al., 1997 [11]). 

To date, sleep-related research and clinical practice has focused on the measurement and improvement of *sleep quality*, even though the construct of sleep quality lacks a rigorous, agreed-upon, definition [4,12]. Rather, sleep quality is described by proxy-quantifiable indices, especially sleep quantity, ease of sleep initiation, ease and degree of sleep maintenance, and frequency of night-time awakenings [12]. Related, sleep research and practice has overwhelmingly focused on sleep problems and pathologies and the use of sleep quality measures, even though most people do not meet clinical criteria for a sleep disorder [5]. 

Over the last decade, there have been calls to approach good sleep in terms of *sleep health and satisfaction* instead of sleep quality, given that sleep quality centres on sleep disturbance and pathology [13]. Buysse (2014, p.12) [13] described good sleep/sleep health as “…a multidimensional pattern of sleep-wakefulness, adapted to individual, social, and environmental demands that promotes physical and mental well-being”. Such a *positive* approach to the experience of good sleep is advantageous. Firstly, it addresses sleep patterns experienced by *all* people, regardless of whether they experience sleep disturbances. Second, it takes a *flexible* approach to people’s understanding of good sleep, acknowledging that it may vary as a function of time and context. Third, it diverts attention away from quantifiable, indices and proxies of good sleep towards *subjective* conceptualizations of good sleep (e.g., one may feel satisfied with only a few hours of sleep, while someone else may feel restless and anxious throughout long hours of sleep and thus unsatisfied). Not only do we agree with this positive approach to the sleep experience, but we argue that it is pertinent, even urgent, given the emergence of *orthosomnia*, i.e., the obsessive preoccupation with experiencing “perfect” sleep, free from disturbances [14]. While currently not a recognised sleep pathology, orthosomnia is emerging as a genuine condition, attributed to sociomedical norms advocating for absolute, measurable, targets of quality sleep (e.g., eight hours of uninterrupted sleep), and exacerbated by consumer sleep technology (e.g., sleep apps and trackers) [15]. Individuals with orthosomnia tend to exhibit insomnia-like symptoms, including difficulty falling asleep, waking up throughout the night, waking up too early, feeling tired, anxious, and not being able concentrate during the day [16,17].

At present, *sleep satisfaction* and *sleep quality* appear to be used interchangeably in the literature [13] and research exploring similarities and differences between the two constructs is extremely limited [18,19]. However, while related, sleep satisfaction and sleep quality are likely different constructs and conflating them is not apt [5,18,19].

Considering these issues, recent empirical efforts [5,18,19] have attempted to disentangle sleep quality and its measurement from sleep satisfaction and its measurement. Ohayon et al., [5,18] published the first sleep satisfaction (instead of sleep quality) scale targeted to the general US population. Still, Ohayon et al., developed this scale without systematically generating a conceptual definition of sleep satisfaction. Instead, they *described* sleep satisfaction as representing “…a positive effect, not merely the absence of a negative effect” [18], p. 7 and proposed “…appropriate sleep satisfaction elements” [18], p. 6, which center on capturing one’s feelings towards one’s sleep. 

We note that despite an emergent interest in sleep satisfaction, there is still a lot of ambiguity regarding its conceptualization, definition, and measurement. A good grasp of the construct of sleep satisfaction is necessary to understand its relevance to the health sector, and ultimately, support population health. The present research builds on and extends Ohayon et al.’s [18] attempt to disentangle sleep satisfaction from sleep quality and is part of a larger project that aims to develop a UK sleep satisfaction measure via community-based participatory research (see Protogerou et al., 2022 [19]). Specifically, the present research is a *rapid review* of studies relating to sleep satisfaction within the context of health. It aims to generate a conceptual definition of sleep satisfaction through a systematic process.

## 2. Materials and Methods

### 2.1. Design

A rapid review approach was used to identify available definitions of sleep satisfaction in a body of literature sampling ‘healthy’ adults, i.e., adults without a diagnosed disease or a sleep disorder. A rapid review is a simplified systematic review that follows all, or most, of the principle steps of a systematic review but omits, or simplifies, some steps to provide an evidence synthesis in a resource-efficient manner (see Moons et al., 2021 [20] for a description of the rapid review approach). A rapid review design was deemed apt for the following reasons. First, a rapid review responds to calls from health systems and stakeholders for accelerated syntheses of timely evidence to inform and support decision-making [21]. The present review was conducted to support our local community partners Suffolk Mind (https://www.suffolkmind.org.uk), a charity that provides mental health services with a focus on sleep health to the Suffolk community (a county located in the East of England, UK). In partnership with Suffolk Mind, we established a need for developing a novel self-report measure to assess sleep satisfaction, targeted to the characteristics of the wider Suffolk community. The present rapid review is a *component* of this larger project. Second, a rapid review may be employed- to update a body of literature [22], and the present rapid review *updates* and *extends* Ohayon et al.’s (2018) [18] literature review on sleep satisfaction and sleep quality. Third, a rapid review may result in a concise yet high-quality evidence synthesis within project constraints when there are finite resources, such as tight timelines, limited personnel, and limited funding [23]. The present review was designed to be completed in a resource-efficient manner: within 8 weeks (between August and September 2022), by the first author, alone.

The present review adhered to emerging rapid review guidelines (i.e., Garritty et al., 2021 [24]; Moons et al., 2021 [20]) to ensure rigorous design, execution and reporting standards. Furthermore, the present manuscript met the PRISMA extension for scoping reviews (PRISMA-ScR: Tricco et al., 2018 [25]) checklist criteria. While the PRISMA-ScR was developed for scoping reviews, we find that it can also be used for rapid reviews, and for the purposes of the present study, we label the extension PRISMA-RR (RR for Rapid Review). The PRISMA-RR, the review protocol and all materials of this rapid review are available at https://osf.io/4wef8/.

### 2.2. Search Strategy and Screening

This rapid review updated and extended Ohayon et al.’s (2018) [18] systematic literature review of potential descriptors and indicators of sleep satisfaction. Ohayon et al. searched the NCBI PubMed database for peer-reviewed original scientific research published from January 1, 2007, to April 1, 2017, in English. The present review followed Ohayon at al.’s search strategy for records published between January 2017 and July 2022 and extended it to include review studies. 

The search terms, with filters, were ((“sleep satisfaction”[All Fields] OR “Sleep Quality”[All Fields] OR “Sleep Quality”[MeSH Terms]) AND (“Personal Satisfaction”[MeSH Terms] OR (“personal”[All Fields] AND “satisfaction”[All Fields]) OR “Personal Satisfaction”[All Fields] OR “satisfaction”[All Fields] OR “satisfactions”[All Fields] OR “satisfaction s”[All Fields] OR “Personal Satisfaction”[MeSH Terms]) AND (“sleep”[MeSH Terms] OR “sleep”[All Fields] OR “sleeping”[All Fields] OR “sleeps”[All Fields] OR “sleep s”[All Fields] OR “sleep/psychology”[MeSH Terms])) AND ((y_5[Filter]) AND (humans[Filter]) AND (english[Filter]) AND (alladult[Filter])). Inclusion and exclusion criteria are presented in Table 1. 

Retrieved records were screened in two stages: title and abstract, and full text (PRISMA flowchart—Figure 1). Records and detailed screening procedures are available at https://osf.io/4wef8/.

### 2.3. Conceptual Definition Generation Procedure 

A conceptual definition of sleep satisfaction was generated based on Podsakoff et al.’s (2016) [26] four-phase process for defining constructs. Briefly, Posdakoff et al.’s conceptual definition process begins with the identification of existing definitions, attributes, antecedents, correlates, consequences, and operationalizations of the construct. Operationalization entails defining the procedures (i.e., “operations”) used to measure *indicators* of the constructs under investigation to “capture” the phenomena or constructs of interest [27]. In phase two, construct attributes, definitions, antecedents, correlates, consequences, and operationalizations identified in phase one are organized into themes, while establishing if construct attributes are necessary (essential) and/or sufficient (unique) properties of the focal construct. These attributes may also be compared to attributes of similar constructs to ensure that the focal construct is distinguishable from similar ones. The specification of necessary attributes is a critical part of the definition generation process, as the necessary attributes will be incorporated in the definition and, ultimately, guide construct measurement. In phase three, a definition is generated, incorporating the necessary construct attributes—and describing its “nature” or “essence”. The definition should state what the construct is, its fundamental antecedents or consequences, the referents to which it applies, its stability over time, and generalizability across situations. Table 2, Table 3, Table 4 and Table 5 depict phases one and two and three. Phase four is described by Podsakoff et al. as an ‘optional’ phase whereby steps are taken (if possible and necessary) to further refine the conceptual definition.

### 2.4. Data Extraction and Analysis

Relevant study information was extracted and organized in tabular form. First, studies were *described* in terms of their publication date, research design, country, participant type, sample size, health focus, correlates and indicators of satisfaction, sleep measures employed, and strength of evidence (Table 2). Then, *factors associated with sleep satisfaction* were situated in categories to which they conceptually “belonged” in reverse order of frequency and prominence across the studies (factors appearing most frequently, first) (Table 3). Then, *operationalization indicators of sleep satisfaction* were situated in categories to which they conceptually “belonged”, in reverse order of frequency and prominence across the studies (indicators appearing most frequently, first) (Table 4). These latter two tasks reflect an *“unconstrained” content analysis* approach, put forth by Elo and Kyngäs (2008) [28]. In line with this approach, data were organized in *unconstrained categorization matrices* that allowed categories to develop inductively using the steps of grouping, categorization and abstraction. 

Descriptive data analyses were conducted with IBM SPSS Statistics (Version 21).

## 3. Results

### 3.1. Descriptive Summary of Study Characteristics

Fifty-one studies qualified for inclusion in the rapid review. Study characteristics and main findings are summarized in Table 2. Studies were published between 2017 and 2022 (median 2020) and employed a mix of designs, with some type of survey design most frequently used (*k* = 38, 73.1%). Studies were conducted across several countries, with the US most frequently represented (*k* = 17, 33.3%), followed by China (*k* = 5, 9.8%); African countries were not represented. Studies sampled young, middle-aged and older adults, across settings (e.g., university, family, health/clinical, home, community), and occupations (e.g., physicians, nurses, police officers, seafarers, remote home office workers), with a sample size of *N* > 218,791. Four studies (7.84%) did not report sample size. Most studies (*k* = 43, 84.3%) utilized a self-reported sleep measure (typically a questionnaire, diary, interview, or a mix of those), and eight (15.7%) used a combination of self-reported and physiological measures (e.g., sleep tracker apps, polysomnography, actigraphy). No study utilized a physiological measure alone. Sleep satisfaction was conceptualized as, measured as, and/or equated to *sleep quality* in more than half of the studies (*k* = 29.6%). Eight studies (15.7%) used a singular item to assess sleep satisfaction directly (e.g., “how satisfied are you with your sleep”), typically scored on a Likert scale. One study measured sleep satisfaction as “sleep experience evoking good feelings” and another study as “desire to change sleep experience”. Only the two studies by Ohayon et al., (2018; 2019) [5,18] used a dedicated sleep satisfaction measure (scale). No conceptual definition of sleep satisfaction appeared in the reviewed studies. 

The strength or ‘level’ of the reviewed evidence was classified using Guyatt et al.’s (1995) [29] hierarchy of evidence system. In this system, levels of evidence are assigned to studies, or other sources of evidence, based on their research design, study quality, and applicability to population care. Levels of evidence range from 1 to 5, indicating risk of bias. Level 1 is assigned to study designs with the lowest likelihood of risk of bias (systematic review or meta-analysis); this is classed as strong evidence. On the other hand, level 5 is assigned to designs with the highest likelihood of risk of bias (single descriptive/survey or qualitative study; this is classed as weak evidence. Lower likelihood of risk of bias is reflected in level 2 designs (randomized controlled trials), moderate risk of bias is reflected in level 3 designs (experiment without randomization), and higher risk of bias is indicated in level 4 designs (observational epidemiologic/case control-cohort studies). 

**Table 2 behavsci-14-00942-t002:** Key study characteristics, listed in alphabetical order by first author.

Study	Study Design	Country ^1^	Population Type	Sample Size (N)	Health-Related Foci	Correlates of Sleep Satisfaction ^2^	Sleep Measure Type	Sleep Satisfaction Operationalization Indicators	Level of Evidence ^3^
Aalto et al. (2018) [30]	Longitudinal survey	Finland	Physicians	1462	Wellbeing	Distress (−); workload (−); good team climate (+); collegial support (+).	Self-report (Jenkins scale).	Sleep onset latency; nocturnal awakenings; waking up before intended; tiredness upon/after awakening.	5
Abraham et al. (2017) [31]	Interview	USA	Older adults	116	Sleep problems	Non-prescription sleep aids (antihistamines, melatonin, analgesics) (+); distress (−); bathroom use (−); non-sleep-related ailments (−); caffeine use (−).	Self-report (interview questions).	Difficulty falling asleep; difficulty staying asleep.	5
Akay et al. (2019) [32]	Longitudinal survey	Germany	General population	76,046	Sleep satisfaction; sleep duration.	Relative income (+); absolute (household) income (+).	Self-report (survey questions).	Sleep satisfaction ≡ sleep quality.	4
Arpin et al. (2018) [33]	Longitudinal survey	USA	Military veterans and their spouses	159	Sleep quality; sleep duration; difficulty falling asleep.	Positive relationship functioning between romantic partners (+).	Self-report (PSQI).	Sleep quality; sleep duration; difficulty falling asleep.	5
Brindle et al. (2018) [34]	Cohort survey	USA	Community-dwelling adults	161	Sleep health; daytime sleepiness.	Childhood trauma (−).	Physiological (wrist actigraphy); self-report (Pittsburgh sleep diary, ESS).	Sleep satisfaction ≡ sleep quality.	4
Chang & Chang (2019) [35]	Cross-sectional survey	Taiwan	Female shift-working nurses	178	Sleep quality	Job satisfaction (+).	Self-report (PSQI).	Sleep quality; sleep latency; sleep duration; sleep efficiency; sleep disturbances; daytime alertness.	5
Cintron et al. (2018) [36]	Randomized controlled trial	USA	Early menopausal women	727	Sleep quality; sleep domains.	Hormonal replacement therapies (HRT) (+).	Self-report (PSQI).	Sleep satisfaction ≡ quality; sleep latency; sleep duration; sleep efficiency; sleep disturbances; daytime alertness/dysfunction.	2
Costa et al. (2022) [37]	Online cross-sectional survey	Italy	Remote home office workers	94	Work performance; mood; sleep quality.	Satisfaction with work performance (+); distressed mood (−); fear of COVID 19 infection (−); perceived negative impact of pandemic on life (−).	Self-report (questionnaire items).	Difficulty falling asleep; nocturnal awakenings with difficulty falling back to sleep; nightmares.	5
Cvejic et al. (2018) [38]	Cross-sectional survey	Australia	Undergraduates	59	Wellbeing; academic performance; functional capacity; sleep quality.	Distress (−); physical health (+).	Self-report (PSQI, sleep diary).	Sleep quality; refreshed upon awakening.	5
Dang et al. (2021) [39]	Cross-sectional survey	China	Community-dwelling older adults	837	Hopelessness; health status; social networks; sleep quality.	Hopelessness (−); social networks (+).	Self-report (PSQI).	Overall sleep quality (composite indicator).	5
Das-Friebel et al. (2020) [40]	Longitudinal survey	UK	Undergraduates	101	Affect; sleep quality.	Bedtime social media use (0); negative affect (−).	Self-report (PSQI).	Sleep duration; sleep quality.	5
De Jonge et al. (2018) [41]	Cohort survey	The Netherlands	Health care workers	203	Recovery from work activities; sleep quality.	Household/childcare care off-job activities (+); leisure off-job activities (−).	Self-report (item from Maastricht Questionnaire).	Difficulty falling asleep.	4
DeSantis et al. (2019) [42]	Longitudinal survey	USA	Community-dwelling adults	738	Distress; body mass index; physical functioning; sleep health.	Physical functioning (+); distress (−).	Self-report (diary question).	Sleep satisfaction.	5
Dohrmann et al. (2020) [43]	Cross-sectional survey	Denmark	Seafarers	193	Work stressors (physical, psychological); fatigue; sleep satisfaction.	Work stressors (−).	Self-report (questionnaire item).	Sleep satisfaction.	5
Dueren et al. (2022) [44]	Evidence Synthesis (Qualitative Systematic Review)	Various	General population	Not reported	Tactile intimacy (sexual, non-sexual); sleep quality.	Sexual touch (+); sexual activity (+)	Self-report (various); physiological (various).	Sleep quality (composite indicator).	1
Furihata et al. (2020) [45]	Cross-sectional survey	Japan	Female nurses	2482	Depression; sleep health.	Depressed mood (−)	Self-report (questionnaire item).	Rested/ refreshed upon awakening.	5
Gillet et al. (2020) [46]	Cross-sectional survey	France	Nurses	378	Job demands (workload, emotional dissonance); relaxation; sleep quality.	Workload (−); presenteeism (−); emotional dissonance (−); emotional exhaustion (−).	Self-report (questionnaire item from PSQI).	Sleep quality.	5
Gu et al. (2019) [47]	Online cross-sectional survey	China	Nurses	2889	Occupational stress; wellbeing; sleep quality.	Occupational stress (−).	Self-report (PSQI).	Sleep quality (composite indicator).	5
Hawkins et al. (2021) [48]	Cohort survey	USA	General population	4837	Weight-loss; sleep health.	Body Mass Index (BMI) (−).	Self-report (doctor-reported sleep disturbances).	Doctor-reported sleep disturbances (indirect indicator of sleep satisfaction).	4
Her & Cho (2021) [49]	Evidence Synthesis (Systematic Review and Meta-analysis)	Various	General population	1657	Health behaviour; Sleep quality.	Aromatherapy (+).	Self-report (various).	Sleep quality (composite indicator).	1
Hidaka et al (2020) [50]	Cohort survey	Japan	General population	49,483	Health behaviours; sleep quality.	Physical activity/exercise (+); eating close to bedtime (−); alcohol use (−).	Self-report (questionnaire item).	Sleep duration; restful sleep.	4
Hinz et al. (2018) [51]	Cohort survey	Germany	Community sample	9711	Satisfaction with life; sleep quality.	Satisfaction with life (+).	Self-report (PSQI, ESS).	Sleep quality (composite indicator); daytime alertness.	4
Hussain et al. (2022) [52]	Cohort survey	Canada	Community-dwelling adults	30,097	Health behaviours; sleep patterns; sleep satisfaction.	Tobacco smoking (−); alcohol use (−).	Self-report (questionnaire items).	Sleep satisfaction.	4
James et al. (2018) [53]	Randomized controlled trial	UK	Police officers	50	Fatigue; sleep quantity; sleep quality.	Fatigue (−).	Physiological (wrist actigraphy); self-report (PSQI, ESS).	Sleep quality.	2
Kang et al. (2020) [54]	Evidence Synthesis (systematic review and meta-analysis)	Various	Shift-work nurses	Not reported	Sleep quality; health-related interventions.	Aromatherapy (+).	Physiological; Self-report.	Sleep quality (composite indicator).	1
Kent et al. (2019) [55]	Cross-sectional survey	USA	Married heterosexual couples	90	Romantic relationship attachment anxiety; emotional avoidance; sleep quality.	Anxious attachment (−); emotional avoidance (−).	Self-report (PSQI).	Sleep quality (composite indicator).	5
Kline et al. (2021) [56]	Controlled trial without randomization	USA	General population	125	Weight-loss; sleep health.	Weight-loss (+); fat loss (+).	Physiological (wrist actigraphy); self-report (PSQI satisfaction item, ESS).	Sleep satisfaction; daytime alertness.	3
Krzych et al. (2019) [57]	Online cross-sectional survey	Poland	Physicians	786	Health behaviours; work conditions; chronic disease; sleep quality.	Tobacco smoking (−); presence of chronic disease (−); more workdays (−); being female (−); being older (−).	Self-report (Sleep Quality Scale—SQS).	Sleep satisfaction; daytime alertness.	5
Kubala et al. (2020) [58]	Cohort survey	USA	General population	114	Physical activity; sleep health.	Moderate-to-vigorous physical activity (+).	Self-report (questionnaire item).	Sleep satisfaction.	4
Lee & Lawson (2021) [59]	Cohort survey	USA	General population	441	Wellbeing; sleep health.	Perceived stress (−); presence of chronic disease (−).	Physiological (wrist actigraphy); self-report (diary, PSQI item).	Sleep satisfaction ≡ quality.	4
Ohayon et al. (2018) [18]	Various (systematic review, expert consensus).	USA	General population	Not reported	All health-related outcomes; sleep satisfaction.	Sleep environment (+); sleep initiation (+); sleep maintenance (+).	Self-report (various).	Sleep satisfaction (as a positive, satisfactory sleep experience), comprising of the following indicators: Feeling good-about one’s own sleep, upon awakening, during the next day, about sleep onset time, about amount of sleep on weekdays and weekends; sleep being affected by bedding, bedroom temperature, noise and light; falling back asleep easily after nocturnal awakening(s); undisturbed sleep; desire to change sleep aspects.	1
Ohayon et al. (2019) [5]	Various (online survey; interviews)	USA	General population	Survey (n = 111); interviews (n = 13)	All health-related outcomes; sleep satisfaction.	Overall health (+); life satisfaction (+); stress (+); experience of sleep problems (−); sleep medication use (−); comfortable bedding (+); being female (−); age (+); living in rural environment (+).	Self-report (SST).	Sleep satisfaction (as a positive, satisfactory sleep experience), comprising the following indicators: overall sleep satisfaction; feeling refreshed upon awakening; feeling alert during the day; sleep efficiency; nocturnal awakenings; ease of falling back asleep after nocturnal awakenings; amount of sleep; ease of achieving a relaxed mental state prior to falling asleep.	5
Pang et al. (2021) [60]	Evidence Synthesis (systematic review and meta-analysis)	Various	General population	555	Neck and spinal issues; sleep quality.	Pillow designs (0).	Self-report (PSQI; sleep diaries).	Sleep quality (composite indicator).	1
Papi & Cheraghi (2021) [61]	Cross-sectional survey	Iran	Older adults	679	Life satisfaction; sleep quality.	Life satisfaction (+).	Self-report (PSQI item).	Sleep quality (composite indicator).	5
Peltz & Rogge (2022) [62]	Longitudinal survey	USA	Parents of adolescents	193	Distress; parent-child relationship; couple relationship; sleep quality.	Distress (−); child’s sleep quality (+)	Self-report (SST).	Sleep satisfaction ≡ quality (composite indicator).	5
Richter et al. (2019) [63]	Cohort survey	Germany	New and experienced parents	4659	Sleep satisfaction; sleep duration.	Childbirth (−); breastfeeding (−); being female (−).	Self-report (questionnaire item).	Sleep satisfaction.	4
Rodriguez-Stanley et al. (2020) [64]	Cohort survey	USA	Married and cohabitating couples	2644	Wellbeing; marital quality; sleep quality.	Perceived fairness of housework distribution (+); household chores hours (0); Socio Economic Status-SES (+).	Self-report (PSQI).	Sleep quality; sleep latency, sleep duration; sleep efficiency, sleep disturbance; daytime dysfunction.	4
Salvi et al. (2020) [65]	Cross-sectional survey	Brazil	Undergraduates	195	Quality of life; eating habits; sleep quality.	Quality of life (+).	Self-report (PSQI).	Sleep quality (composite indicator).	5
Seol et al. (2021) [66]	Randomized controlled trial	Japan	Older adults	60	Exercise timing; sleep parameters; sleep satisfaction.	Exercise during the evening (+).	Physiological (wrist actigraphy); self-report (PSQI).	Sleep satisfaction ≡ quality (composite indicator)	3
Son et al. (2020) [67]	Cross-sectional survey	South Korea	General population	332	Shoulder, head and neck pain and fatigue; sleep quality.	Head and neck fatigue (−); shoulder pain (−); pillow comfort (−).	Self-report (PSQI).	Sleep quality (composite indicator).	5
Štefan et al. (2018) [68]	Cross-sectional survey	Croatia	Undergraduates	2100	Physical activity; sleep quality.	Physical activity/exercise (+).	Self-report (PSQI).	Sleep quality; sleep latency, sleep duration; sleep efficiency, sleep disturbance; daytime dysfunction; composite sleep quality indicator.	5
Targa et al. (2021) [69]	Online cross-sectional survey	Spain	General population	71	Mood; sleep health; sleep quality.	Positive mood (+).	Self-report (PSQI).	Sleep quality; sleep latency, sleep duration; sleep efficiency, sleep disturbance; daytime dysfunction; composite sleep quality indicator.	5
Tavernier et al. (2019) [70]	Longitudinal survey	USA	Undergraduates	154	Basic psychological needs; sleep quality.	Perceived fulfilment of basic psychological needs (+).	Self-report (PSQI item).	Sleep quality.	5
Toussaint et al. (2020) [71]	Cohort survey	USA	General population	1423	Forgiveness of others; self-forgiveness; distress; life satisfaction; physical health; sleep quality.	Forgiveness of others (+); self-forgiveness (+); distress (−); life satisfaction (+); physical health (+).	Self-report (PSQI item).	Sleep quality.	4
Varghese et al. (2020) [72]	Cross-sectional survey	Italy	General population	3120	Sleep dissatisfaction; sleep duration; sleep quality.	Age (−); SES (−); being female (−); divorce/separation (−); living with children (+); living with pets (−).	Self-report (PSQI item).	Sleep quality.	5
Wang & Boros (2020) [73]	Randomized controlled trial	Hungary	General population	54	Daily exercise; stress; life satisfaction; sleep quality.	Daily exercise (+).	Self-report (PSQI).	Sleep quality; sleep latency, sleep duration; sleep efficiency, sleep disturbance; daytime dysfunction; composite sleep quality indicator.	3
Wang et al. (2019) [74]	Cross-sectional survey	China	Undergraduates	6284	Suicidal ideation; mood; lifestyle; sleep quality.	Suicidal ideation (−).	Self-report (PSQI).	Sleep quality; sleep latency, sleep duration; sleep efficiency, sleep disturbance; daytime dysfunction; composite sleep quality indicator.	5
Yorgason et al. (2018) [75]	Cohort survey	USA	Older married couples	191	Marital relationship quality; mood; sleep quality.	Positive marital events (+); marital satisfaction (+); positive mood (+); being female (−).	Self-report (questionnaire item).	Sleep quality.	4
Yuan et al. (2019) [76]	Cross-sectional survey	China	Nurses	923	Work-related conditions; sleep quality.	Shift work (−); job demands (−); exposure to environmental work hazards (−); fatigue (−); job satisfaction (+); supportive relationships at work (+).	Self-report (PSQI).	Sleep quality (composite indicator).	5
Zandy et al. (2020) [77]	Cohort survey	Canada	General population	10,806	Tobacco smoke exposure; sleep quality.	Tobacco smoke exposure (−); being female (−).	Self-report (questionnaire item).	Sleep satisfaction ≡ sleep quality (sleep satisfaction as refreshed sleep).	4
Zheng et al. (2019) [78]	Quasi-experimental	China	Undergraduates	10	Sleep quality.	High temperature weather (−).	Physiological (non-wearable sleep monitoring belt); self-report (questionnaire item).	Sleep satisfaction ≡ sleep quality.	3

Note. Abbreviations. PSQI = Pittsburgh Sleep Quality Index [79]. ESS = Epworth Sleepiness Scale [10]. SQS = Sleep Quality Scale [80]. SST = Sleep Satisfaction Tool [18]. ^1^ Where data were collected. ^2^ A positive sign (+) indicates a positive association/linkage with sleep satisfaction, a negative sign (−) indicates a negative association/linkage, and zero (0) indicates no association. ^3^ Levels of Evidence Pyramid [29]: Level 1 (source of evidence with lowest likelihood of risk of bias) = systematic review or meta-analysis; Level 2 = Randomized Controlled Trial (RCT); Level 3 = controlled trial without randomization; Level 4 = observational epidemiologic study/case control-cohort study; Level 5 (source of evidence with highest likelihood of risk of bias) = single descriptive/survey or qualitative study.

In the present review, about half of the studies (*k* = 26.5 %) employed single descriptive surveys or qualitative designs, and 14 studies (27.5 %) employed epidemiologic, case-control and cohort designs; thus, the evidence generated from these studies is classified as weak. Six studies (11.7%) provided stronger evidence, from experimental designs. Five studies (9.5%) generated strong evidence as they were systematic reviews. Table 2 also provides a snapshot of correlates and operationalization indicators of sleep satisfaction, which are fleshed out in the next two sections. 

### 3.2. Phase 1 of Definition Generation Process: Identifying Factors (Correlates, Antecedents, Consequences) Associated with Sleep Satisfaction

Content analysis revealed six main categories of factors associated with sleep satisfaction (i.e., factors associated with enhanced or diminished sleep satisfaction, via tests of correlation or prediction or change). Table 3 provides the categories along with factor frequency and prominence within studies. Prominence of identified factors was gauged by co-occurrence, that is, the frequency with which the factors was associated with sleep satisfaction across included studies. 

**Table 3 behavsci-14-00942-t003:** Categorization matrix: Emerging categories and frequency and co-occurrence of factors associated with sleep satisfaction, situated in reverse order of factor frequency (factors appearing most frequently, first).

Sleep Context/Ambiance Co-Occurrence: 8	Interpersonal Relationships Co-Occurrence: 19	Health States Co-Occurrence: 37	Health Behaviours Co-Occurrence: 16	Individual-Level Characteristics Co-Occurrence: 11	Life Satisfaction/Standard of Living Co-Occurrence: 15
Sleeping with a comfortable pillow (+) (**3)**	Experiencing sexual activity and sexual touch (+) **(2)**	Having positive mood/affect (+) **(16)**	Exercising/being physically active (+) **(5)**	Being female (−) **(5)**	Satisfied with job (+) **(4)**
Being affected by temperature (−) **(2)**	Having collegial support (+) **(2)**	Being healthy and well (+) **(3)**	Taking non-prescription sleep aids (+) (supplements, aromatherapies) **(3)**	Being older (+) **(2)**	Satisfied with life (+) **(4)**
Being affected by light (−) **(1)**	Having positive relationship with romantic partner (+) **(2)**	Being fatigued (−) **(3)**	Using alcohol (−) **(2)**	Being younger (+) **(1)**	High workload (−) **(3)**
Being affected by noise (−) **(1)**	Having experienced childhood trauma (−) **(2)**	Experiencing bodily pain (−) **(2)**	Smoking tobacco (−) **(2)**	Having higher socioeconomic status (+) **(2)**	Exposed to environmental hazards or toxicants (−) **(2)**
Sleeping in rural environment (+) **(1)**	Having social networks (+) **(2)**	Suffering from chronic diseases (−) **(2)**	Losing weight and fat (+) **(2)**	Having higher income (+) **(1)**	
	Caring for/living with children (+) **(2)**	Experiencing menopause (−) **(1)**	Eating ‘healthily’ (+) **(1)**		Overall quality of life (+) **(1)**
	Having good work team (+) **(1)**	Functioning well physically (+) **(1)**	Taking hormonal replacement therapy (+) (HRT) **(1)**		Working shifts (−) **(1)**
	Child sleep quality (parent-child concordance) (+) **(1)**	Engaging in presenteeism (−) **(1)**			
	Perceiving that housework is distributed fairly with spouse (+) **(1)**	Fearing a COVID-9 infection (−) **(1)**			
	Forgiving of others (+) **(1)**	Fear of COVID-19 pandemic’s impact on life (−) **(1)**			
	Being divorced (−) **(1)**	Experiencing sleep problems (−) **(1)**			
	Living with/caring for pets (−) **(1)**	Recently experiencing childbirth (−) **(1)**			
	Being satisfied in marriage (+) **(1)**	Breastfeeding (−) **(1)**			
		Perceiving that psychological needs are met (+) **(1)**			
		Forgiveness of self (+) **(1)**			
		Engaging in negative thinking patterns (−) **(1)**			

Note. Numbers in parentheses show the *k* of studies in which a factor appears, indicating factor frequency. Co-occurrence values show how many times a factor appears across studies, indicating factor prominence. A positive sign (+) denotes a statistically significant positive association (correlation or prediction or change) with sleep satisfaction and a negative sign (−) denotes a statistically significant negative association with sleep satisfaction.

*Health states* (relating to body and mind) appeared most prominently across the sample of studies (*k* = 37, 72.5%). Health states *positively* associated with sleep satisfaction included: having positive mood/affect, being healthy and well, functioning well physically, and believing that psychological needs are met. Health states *negatively* associated with sleep satisfaction included: experiencing fatigue, bodily pain, and chronic disease. States such as experiencing menopause, childbirth and breastfeeding, were also negatively associated with sleep satisfaction, but were examined to a lesser degree.

Factors concerning *interpersonal relationships* appeared somewhat prominently in the studies (*k* = 19, 37.2%). Interpersonal factors *positively* associated with sleep satisfaction included: having collegial support and good teamwork, having a social network, enjoying a satisfying romantic or marriage relationship, experiencing sexual activity and touch, perceiving that housework is distributed fairly with spouse, caring for/living with children, and with children that are good sleepers, and being forgiving towards others. Interpersonal factors *negatively* associated with sleep satisfaction included: having experienced childhood trauma, being divorced, and living with/caring for pets. 

*Life satisfaction and standard of living factors* were examined in 15 studies (29.4%). Being satisfied with life and work, perceiving having good quality of life, and having higher income and higher socioeconomic status were factors associated with enhanced sleep satisfaction. However, having a high workload, working shifts, and being exposed to environmental hazards or toxicants were all associated with reduced sleep satisfaction.

Linkages between sleep satisfaction and *health behaviours* appeared in sixteen studies (31.4%). Exercising and being physically active, losing weight and fat, taking nonprescription supplements to support sleep, eating healthily and taking hormonal replacement therapy if in menopause, were all associated with *enhanced* sleep satisfaction. Dring alcohol and smoking tobacco were associated with *reduced* sleep satisfaction. 

*Individual-level characteristics* appeared in eleven studies (21.5%). Being female was associated with *less* sleep satisfaction, having higher socioeconomic status or higher income was associated with enhanced sleep satisfaction, and age was associated with enhanced sleep satisfaction, albeit inconsistently (both being younger and being older were positively associated with sleep satisfaction, in different studies).

Finally, *sleep ambiance* appeared as a factor category in eight studies (15.7%). Sleeping with a comfortable pillow and in a rural environment were associated with enhanced sleep satisfaction, while being affected/disturbed by temperature, light and noise were associated with *diminished* sleep satisfaction.

### 3.3. Phase 2 of Definition Generation Process: Identifying Operationalizations of Sleep Satisfaction

Content analyses elucidated the ways sleep satisfaction was *operationalized* the reviewed studies. Four categories of operationalization indicator emerged. Table 4 provides operationalization indicators of sleep satisfaction in reverse order of prominence, defined as the frequency with which an indicator appeared across studies. 

**Table 4 behavsci-14-00942-t004:** Categorization matrix: Emerging categories and frequency and co-occurrence of sleep satisfaction operationalization indicators, situated in reverse order of frequency and measures.

Pre-Sleep Experience ^1^ Co-Occurrence: 15	Amid-Sleep Experience ^1^ Co-Occurrence: 36	Post-Sleep Experience ^1^ Co-Occurrence: 22	Sleep Ambiance/Context Co-Occurrence: 4	Sleep Satisfaction vs. Sleep Quality Measure
Sleep onset latency **(8)**	Sleep amount/ duration **(12)**	Daytime alertness **(13)**	Bedding comfort **(1)**	Sleep quality **(23)**
Difficulty falling asleep **(6)**	Sleep efficiency **(9)**	Feeling refreshed/rested after awakening **(6)**	Bedroom comfort—light **(1)**	Sleep satisfaction **(8)**
Ease of achieving a relaxed mental state before falling asleep (**1)**	Sleep disturbances **(8)**	Waking up earlier than intended **(1)**	Bedroom comfort—noise **(1)**	Sleep quality **≡** sleep satisfaction ^2^ **(7)**
	Nocturnal awakening(s) **(3)**	Desire to change sleep experience **(1)**	Bedroom comfort—temperature **(1)**	
	Ease of resuming sleep after nocturnal awakening(s) **(3)**	Good feelings after sleep **(1)**		
	Sleep experience evoking good feelings **(1)**			
	Nightmares **(1)**			

Note. Numbers in parentheses show the *k* of studies in which indicators appear, suggesting indicator frequency. Co-occurrence values show how many times an indicator appears across studies, suggesting indicator prominence. ^1^ Proxy quantifiable measures of sleep experience. ^2^ Sleep quality and sleep satisfaction were operationalized/treated as identical in the study.

*Proxy quantifiable measures* relating to one’s experience pre, amid, and post sleep were the most prominent indicators of sleep satisfaction. Sleep satisfaction was most frequently operationalized by indicators of one’s *amid-sleep* experience, especially sleep duration (total hours of sleep), sleep efficiency (the ratio of total hours being asleep compared with the hours spent in bed), and sleep disturbances (e.g., coughing, snoring, nocturnal urgency). The pervasive use of proxy quantifiable measures to gauge sleep satisfaction could be explained by the fact that more than half of reviewed studies operationalized sleep satisfaction as sleep quality or treated sleep satisfaction and sleep quality as conceptually identical. Also, indicators relating to one’s sleep environment or *ambiance* were used to operationalize sleep satisfaction (e.g., pillow, bedding, and bedroom comfort).

### 3.4. Phase 3 of Definition Generation Process: Identifying Necessary and Sufficient Attributes of Sleep Satisfaction 

Attributes of sleep satisfaction, i.e., correlates, operationalization indicators, and measures, identified in the content analyses were organized into their necessary and sufficient properties, and in relation sleep quality (Table 5). The necessary attributes were incorporated in the definition of sleep satisfaction and helped distinguish sleep satisfaction from sleep quality. We determined that *the* necessary attribute of sleep satisfaction was *perceived general contentment* with sleep. Furthermore, we understood *health* (physical and mental) to be a strong indicator of sleep satisfaction, and that sleep satisfaction varies with *individual-level* and *contextual* characteristics. 

**Table 5 behavsci-14-00942-t005:** Identifying Necessary and Sufficient Attributes of the Construct of Sleep Satisfaction in Relation to Sleep Quality.

Attributes	Sleep Satisfaction	Sleep Quality	Conclusions
A1. Sleep environment/context characteristics.	Present	Absent	**Necessary ^b^**
A2. Reports contentment with general sleep experience.	Present	Absent	**Necessary**
A3. Reports proxy quantifiable indices of good sleep.	Present	Present	Sufficient
A4. Reports physical and mental health.	Present	Present	**Necessary ^a^**
A5. Reports engaging in health behaviours.	Present	Absent	Sufficient
A6. Reports having positive interpersonal relationships.	Present	Absent	Sufficient
A7. Reports satisfaction with one’s life and standard of living.	Present	Present	Sufficient
A8. Individual-level characteristics.	Present	Absent	**Necessary ^b^**

Note. Necessary attributes are considered focal and included in the definition of sleep quality. ^a^ As an indicator of sleep satisfaction. ^b^ As a covariate of sleep satisfaction.

### 3.5. Phase Four of Definition Generation Process: Formulating the Definition of Sleep Satisfaction 

We reiterate that a conceptual definition of sleep satisfaction was developed based on identifying/analyzing (a) factors associated with sleep satisfaction and operationalizations of sleep satisfaction (Table 3 and Table 4); (b) sleep measurement scale content (Table 1, Table 4 and Table 5); and (c) necessary and sufficient attributes of sleep satisfaction in relation to sleep quality (Table 5). We also reiterate that a construct definition should include what the construct is (its “essence”), its fundamental antecedents and consequences, the referents to which it applies, its stability over time, and its generalizability across situations. 

Based on the above, we propose that sleep satisfaction is an indicator of general health, which can be defined as a personal, introspective, and global judgment about one’s feelings of contentment with one’s sleep, at a particular point in time. Sleep satisfaction is impacted by and varies with sleep environment and individual-level characteristics. The reviewed literature pointed to age (stage of life), gender and socioeconomic status as particularly relevant individual-level characteristics, and to characteristics of one’s bed and bedroom as particularly relevant environmental characteristics.

## 4. Discussion

The purpose of the present rapid review was to generate a conceptual definition of sleep satisfaction. The review synthesized information from fifty-one studies (*N* > 218,791) of various design, sampling adults without a diagnosed medical condition, across a variety of countries, occupations and settings. The definition was generated systematically following the procedure put forth by Podsakoff et al. [26], while the data were content analysed following procedures put forth by Elo and Kyngäs [28].

What stood out the most was the complete absence of a dedicated, fit-for-purpose definition of the construct of sleep satisfaction. A related-issue was the use and measurement of sleep satisfaction and sleep quality as identical constructs. To date, neither sleep satisfaction, nor sleep quality, have been defined on the basis of a systematic definition-generating processes. Notably, sleep quality and sleep satisfaction measurement scales have been developed and widely used in research and clinical practice, in the absence of established, agreed-upon definitions of said constructs. Therefore, our systematic approach to generating a conceptual definition of sleep satisfaction, in relation to sleep quality, is timely and necessary. 

### 4.1. Definition-Generation Process

An essential part of the process of conceptually defining a construct is identifying its ‘antecedents, consequences, or correlates’ to help clarify its meaning [26,81]. In the present review, being satisfied with sleep was prominently associated with better health status, including experiencing positive mood, good physical functioning, and reduced, or absence of, fatigue, pain and disease. Being satisfied with sleep was also associated with engaging in health behaviours (e.g., exercising, abstaining from alcohol and tobacco), having good interpersonal relationships, and being satisfied with life and work. Sleep satisfaction was contingent on contextual (e.g., bedding and bedroom comfort), and individual-level characteristics (age, gender, socioeconomic status). Specifically, women tended to report less sleep satisfaction, while younger and older adults (depending on study) and those of high socioeconomic status, tended to report more sleep satisfaction. Sleeping in a comfortable bed, in the absence of disturbances caused by noise and light, also appeared to improve one’s satisfaction with sleep (Table 3). 

Another part of the process of defining a construct is examining the construct’s measures and operationalization indicators, as well as those of closely related constructs [26,27]. This process is particularly important when there are no available conceptual definitions or when definitions are unclear. Examining measures and operationalizations helps reveal the meaning, or “essence”, of the focal construct, the ambiguities of the focal construct, as well as potential conceptual overlaps with closely related constructs. *Conceptual overlap*, also known as *construct proliferation*, is the development of constructs that have different names but are theoretically or empirically indistinguishable or overlapping in conceptual domains [82]. The present review found that, overwhelmingly, sleep satisfaction, as well as the closely related construct of sleep quality, were operationalized in terms of proxy quantifiable indices, measured by self-report scales (Table 4). Total hours of one’s sleep, the hours spent sleeping compared to the hours staying awake in bed, and estimations of sleep disturbance rates (e.g., frequency of coughing, snoring, waking up to urinate) were the most prominent measures of sleep satisfaction and quality. Also, bedroom characteristics conducive to a good night’s sleep (e.g., having comfortable bedding and bed, not being disturbed by noise and light) were measures of sleep satisfaction but not sleep quality, in a small number of studies. More than half of the reviewed studies operationalized sleep satisfaction as sleep quality, or treated sleep satisfaction and sleep quality as conceptually identical. Evidently, there was extensive construct proliferation between sleep satisfaction and sleep quality in the reviewed literature, which is problematic for several reasons. First, construct proliferation undermines discriminant validity (i.e., the extent to which a construct is distinct from another), as identical or conceptually overlapping constructs are likely treated as distinct and likely resulting in construct redundancy and multicollinearity. This would mean that conclusions on relationships between tested constructs are likely incorrect. Construct proliferation also undermines nomological validity (i.e., the degree to which a construct behaves as it should within a system of related constructs, referred to as a ‘nomological network’), making it difficult to identify related constructs or specify whether the constructs are antecedents, consequences, or correlates of the focal construct. Also, construct proliferation poses risks to construct validity (i.e., the degree to which test is measuring what it claims to measure), increasing the likelihood of a mismatch between constructs and their measures or manipulations. These threats to validity introduced by construct proliferation further suggest that the measures or manipulations of the focal construct contain systematic or random measurement error or both [83]. *The* solution to construct proliferation and its threats to validity and measurement accuracy is to ensure that the constructs are well defined *before* they are measured and manipulated. Unclear, ambiguous, construct definitions tend to only provide general, or, approximate, information about the nature of a construct, and little guidance in terms of its operationalization and measurement [84]. Notably, the present review found that sleep studies have not undertaken any systematic steps to conceptually define sleep satisfaction, in relation to sleep quality, and have measured/tested sleep satisfaction in the absence of a dedicated, fit-for-purpose definition, often conflating it with sleep quality. Similarly, currently used sleep satisfaction and quality scales have been developed in the absence of an analysis of construct properties and definition.

It is important note that the development of measurement scales and the testing of constructs in the absence of conceptual definitions is not specific to sleep research but is common across behavioural and social science research [85,86]. To date, the development of measurement scales with high reliability and good factor structure has been prioritized (at least in quantitative research), while the conceptual analysis and definition of constructs has been downplayed or entirely overlooked [26,87]. 

The last step in our definition generation process was organizing key attributes of sleep satisfaction in terms of whether those reflected necessary or sufficient properties of sleep satisfaction, in relation to sleep quality (Table 5). This was yet another endeavour to distil the essence of sleep satisfaction, while disentangling it from sleep quality. After this, we formulated a definition of sleep satisfaction as *a personal, introspective, and global judgment about one’s feelings of contentment with one’s sleep, at a particular point in time*. Based on the evidence, we also understood that sleep satisfaction is a consistent indicator of general health, impacted by and varying with sleep context and individual-related characteristics. The evidence points to gender, age and socioeconomic status as individual-level characteristics markedly shaping one’s perception of sleep satisfaction. The evidence also points to characteristics of one’s bed, bedding, and bedroom ambiance as contextual characteristics impacting one’s satisfaction with sleep. 

We also identified ‘sufficient’ attributes of sleep satisfaction, i.e., attributes that contribute to, or have relevance to, sleep satisfaction but are not ‘required’ for one to perceive their sleep as satisfactory. Proxy-quantifiable indices of good sleep (e.g., sleep quantity, latency, efficiency), engaging in health behaviours (e.g., abstinence from smoking and drinking, exercising), maintaining positive interpersonal relationships, and reporting good quality of life, were deemed ‘sufficient’ but not necessary attributes and were, thus, not incorporated in the proposed definition. While we would like this to be confirmed by research, we posit that the sufficient attributes may be absent from an individual’s life at a particular point of time, but they may still report experiencing sleep satisfaction. Of these sufficient attributes, self-reported proxy-quantifiable indices of sleep are emblematic of sleep quality and should best be detached from sleep satisfaction.

### 4.2. Implications for Measurement

The present rapid review has strong implications for-sleep satisfaction measurement. In line with our proposed definition, we argue that sleep satisfaction could be measured as a *global* judgment about one’s *feelings of contentment* with one’s sleep, *at a particular point in time*. While further research is needed to generate a measure in line with this definition, we put forth the following recommendations for measurement. 

Sleep satisfaction may be measured with a global, single-item scale, whereby respondents are asked to think of their current sleep and rate their feelings of contentment with their sleep, overall. The item could be scored on a Likert-type scale or a visual analog scale. For example, the item could be phrased as: *considering your sleep experience overall, how content are you with your sleep?* If opted for, a visual analogue scale could be generated from http://www.vasgenerator.net/, accessed on 30 September 2024 developed by Reips and Funke [88].

It is important to note that single-item scales are used in sleep research. In the present review, all studies that reported measuring sleep satisfaction, instead of sleep quality, used a single-item scale that asked-respondents to rate their satisfaction with sleep, typically scored on a Likert scale. Related, Snyder et al. [89] developed the *single-item sleep quality scale (SQS)*, which seeks a self-reported global rating of sleep quality over a 7-day recall period, using a visual analogue scale.

Single-items scales, with comparable or favourable measurement characteristics to multiple-item scales, are also used across areas of behavioural health (for examples of such scales see Di et al. [90]; Smith et al. [91]; and Hoeppner et al. [92]). The brevity and reduced demands on respondents, researchers and clinicians are clear practical benefits of global, single-item scales [93]. 

Additional concerns can be raised about the widely used self-reported proxy-quantifiable scales in terms of their ability to accurately assess sleep. Quintessential proxy-quantifiable measures of sleep include quantity (hours of sleep), latency (the minutes needed to fall asleep), efficiency (ratio of hours spent sleeping compared to the hours staying in bed in percentage), and awakenings (number of times waking up during the night). These would measure *good* sleep in terms of it lasting between six and eight hours; of one being able to fall asleep quickly, typically within 30 minutes; of one being asleep for above 85% of time lying in bed; and one not being awakened or being minimally awakened during sleep. Respondents are typically required to self-report on these measures as occurring “in the last month” (for examples of such measures see Buysse et al., 1989 [79]; Soldatos et al., 2000 [94]; and Yi et al., 2006 [80]). The inherent difficulties in self-estimating with precision such sleep indicators is evident given recall bias, self-report bias, as well as responder and administrator fatigue- all common biases of self-reported measurement (see also Ibáñez et al., 2018 [7]). Accurately self-estimating proxy-quantifiable indices may also be hampered by *sleep state misperception* [95]. People who misperceive their sleep may self-report frequent awakenings and only a few hours of sleep during the night, even though their sleep is normal, as evidenced by their physiological measures [96,97].

A relevant, albeit contentious, discourse relates to whether currently used quantifiable indices of good sleep, especially dictums on the necessity of “eight hours of uninterrupted sleep”, are in line with human evolution or are a recent social construction. Historical accounts of human pre-industrial sleep describe typical sleep as occurring in two distinct phases, bridged by an hour or more of wakefulness [98,99]. Related, experimental studies [100,101] have shown bi-phasic nighttime sleep patterns to naturally emerge in contexts where people are deprived from stimuli affecting sleep (e.g., artificial light, modern technology). 

The impetus to achieve and measure good sleep, as evidenced by consumer sleep tracking technology has also been found to promote orthosomnia, which is an obsessive preoccupation with experiencing “perfect” sleep [14]. 

Lastly, we note that quantifiable sleep measures are not always better able to predict health outcomes (see Kohyama, 2021 [102] for a review), and that subjective perceptions of sleep can impact health outcomes in a profound way, overriding quantifiable indicators of sleep. For example, Draganich and Erdal (2014) [103] constructed false sleep perceptions by telling normal sleepers that they had “below average” or “above average” sleep, based on purported measures of brainwave activity by electroencephalogram. These constructed perceptions (i.e., placebo effects) impacted people’s performance on memory and attention tests on the next day, so that those who were told had experienced “below average” sleep the night before performed worse on cognitive tasks, and those who were told had experienced “above average” sleep the night before performed better. 

Based on the above, we surmise that when it comes to measuring sleep satisfaction among general, non-clinical populations, a shift away self-reported proxy-quantifiable indices, towards a global perceived measure of sleep is apt. On the other hand, physiological, as well as proxy-quantifiable measures of sleep have relevance for populations experiencing sleep disorders and for the assessment of specific dimensions of sleep.

### 4.3. Implications for Research and Practice

Our findings have strong implications for research and practice. Our definition of sleep satisfaction can be applied, as is, in sleep research- to conceptualize, measure and investigate good, satisfactory, sleep. Our definition of sleep satisfaction as a global judgment of one’s contentment with sleep ‘translates’ into measuring sleep satisfaction with a global, singular, self-reported item. The adoption of such a measure has benefits for practice, as it would be quick and easy to administer and score, without the need for administrator training, and with minimal burden on respondents. Still, we strongly recommend further research into developing this singular measure and testing it against other relevant multiple-item measures. 

Our findings indicate that, to date, research on sleep satisfaction and the conceptually related construct of sleep quality, has been conducted in the absence of any systematic attempt to define said constructs. We strongly recommend that sleep studies incorporate-conceptual analyses and definition(s) of the sleep construct(s) manipulated. To that end, our work presents a systematic process of developing conceptual definitions via a rapid review approach.

Broadly, our findings demonstrate that, compared to a full-scale systematic review, a rapid review approach can be instrumental in determining the current state of knowledge to conceptually define constructs, with less burden on researchers. We therefore strongly recommend that rapid reviews are utilized to generate construct definitions.

Lastly, we note that procedures of conceptual analysis and construct definition generation are not typically part of research courses at higher education, and the development of (good) conceptual definitions is unlikely to be part of scholars’ research repertoires. As the absence of conceptual analysis and apt definitions may limit validity, we strongly advocate for incorporating construct generation approaches, like the one demonstrated in the present rapid review, in research curricula and practice.

### 4.4. Strengths and Limitations

This rapid review advances understanding of the construct of sleep satisfaction, by providing a dedicated, fit-for-purpose, definition of sleep satisfaction, with implications for measurement, research, and practice. We followed Podsakoff et al.’s (2016) [26] systematic process for developing conceptual definitions, which included (1) identification of existing definitions, attributes, correlates, consequences, and operationalizations of sleep satisfaction; (2) organization of construct attributes, definitions, correlates, consequences, and operationalizations into themes; (3) specification of the necessary (essential) and sufficient (unique) attributes of sleep satisfaction in relation to sleep quality; and (4) formulation of a sleep satisfaction definition in terms of its essential attributes, the referents to which it applies, its stability over time, and generalizability across situations. Important in our process was attempting to distinguish-sleep satisfaction from the related construct of sleep quality. To our knowledge, this is the first attempt to systematically define sleep satisfaction and disentangle it from sleep quality, filling an evidence gap. 

Furthermore, while *rapid*, the present review employed a rigorous systematic review design, including protocol registration and workflow tracking within the Open Science Framework platform (https://osf.io/4wef8/), and PRISMA flowchart. The review-adhered to emerging rapid review guidelines and rationales for conducting a rapid review (e.g., Moons et al., 2021 [20]). The present manuscript also met the PRISMA extension for scoping reviews (PRISMA-ScR) checklist criteria, adapted for a rapid review (PRISMA-RR). 

However, several limitations should be acknowledged. First, to meet project-specific resources and timelines, only the first author conducted title and abstract screening, data extraction, and data analyses. While this is in line with rapid review approaches [20,23], and co-authors reviewed procedures undertaken, there is the potential that this may have introduced bias in the review. 

A formal appraisal of the quality of the included evidence was not conducted, but a “snapshot” of the strength of evidence was obtained using Guyatt et al.’s (1995) [29] hierarchy of evidence system. Based on this, only 11 studies (21.5%) employed designs that are expected to provide strong, credible evidence (i.e., experimental, systematic reviews and meta-analyses), suggesting that our findings, too, need to be approached with caution. 

Another limitation of the included evidence was the construct proliferation between sleep satisfaction and sleep quality. Only a small number of studies had reported measuring or testing sleep satisfaction itself, and even those studies had not defined sleep satisfaction, systematically. While we took steps to disentangle sleep satisfaction from sleep quality, our findings are still affected by the conceptual overlap between sleep satisfaction and quality in the included studies. It is still difficult to conclude with a high degree of certainty that the construct of sleep satisfaction is unique, and further research is needed to establish our conceptual definition. 

Lastly, we note that construct definition generation processes are necessarily intuitive, relying on the analytic, critical and creative skills of those developing the definition [104]. This reliance on one’s intellectual skills may be seen as a potential limitation of definition generation endeavours. Still, in the present rapid review, the intuitive, creative dimensions of the definition generation process were combined with the systematic, structured, process described above.

## 5. Conclusions

This rapid review filled an evidence gap by systematically defining the construct of sleep satisfaction, in relation to sleep quality. We propose that sleep satisfaction is an indicator of general health, defined as a personal, introspective, and global judgment about one’s feelings of contentment with one’s sleep, at a particular point in time. Furthermore, sleep satisfaction is impacted by and varies as a function of one’s sleep environment and individual-level characteristics. This definition implies that sleep satisfaction may be measured by a singular item whereby one rates their feelings of contentment with sleep, overall. Based on these findings, we recommend further research into developing this sleep satisfaction measure and psychometrically evaluating it. We also recommend using rapid review approaches to conceptually define constructs.

## Figures and Tables

**Figure 1 behavsci-14-00942-f001:**
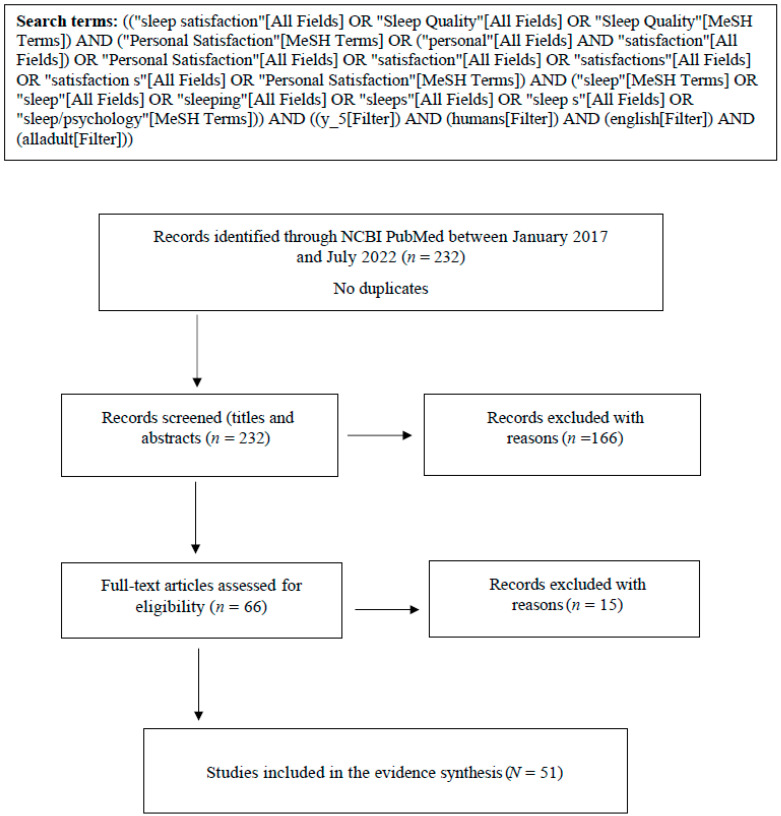
PRISMA flowchart of study inclusion and reasons for exclusion. Figure 1. PRISMA flowchart of study inclusion. Reasons for excluding studies are as follows: animal studies; non-adult populations (<18 years of age); populations with sleep or non-sleep diseases/conditions; studies using objective sleep measures only; settings comprised of special populations (e.g., hospitals, care centers with specialized patient groups, prisons); study types that are unpublished, preprints, conference reports, student theses and dissertations, abstract-only, letters, opinion pieces, protocols, and translations of extant sleep scales; non-English language articles; sleep measures not operationalized.

**Table 1 behavsci-14-00942-t001:** Inclusion and exclusion criteria.

Inclusion	Exclusion
All health-related outcomes; humans; adults, general populations; subjective sleep measures, sleep satisfaction/quality; all settings that are not excluded; all study types that are not excluded; peer-reviewed publications; English language; published between January 2017 and July 2022.	Animal studies; non-adult populations (<18 years of age); populations with sleep or non-sleep diseases/conditions; studies using objective sleep measures only; special-population settings—(e.g., hospitals, care centers with specialized patient groups, prisons); study types that are preprints, conference reports, student theses and dissertations, abstract-only, letters, opinion pieces, protocols, and translations of extant sleep scales; non-English language articles; sleep measures not operationalized.

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
