# Peer review of "Conceptualizing Sleep Satisfaction: A Rapid Review"

_behavsci, 2024, doi:10.3390/bs14100942_

Round 1
Reviewer 1 Report
Comments and Suggestions for Authors
This review aims to introduce the term "sleep satisfaction" and its distinction from "sleep quality". The data analyzed are comprehensive. Overall the manuscript is well-written. I have only a few issues.
The concept of sleep satisfaction is certainly important, especially given the emergence of "orthosomia" as mentioned by the authors. However, objective sleep quality measures are also necessary and sufficient attributes of sleep satisfaction. In Table 5, the few attribues unique to "sleep satisfaction", such as sleep environment, appear vague terms. The authors could elaborate more on what is the true biological/psycohological mechanism underlying the distinction betweeen "sleep satisfaction" and "sleep quality". In other words, are those attributes truely necessary or just arise due to pure correlation?
The authors should perform a through proofreading of the manuscript. A few mistakes in grammar is noted, for example:
1. Page 17 line 1: missing "in".
2. Page 17 line 26: missing "to".
3. Page 18: Table 5/A5: "health behaviour" should be "healthy behavior"?
Author Response
REVIEWER’S COMMENT: This review aims to introduce the term "sleep satisfaction" and its distinction from "sleep quality". The data analyzed are comprehensive. Overall, the manuscript is well-written. I have only a few issues.
AUTHORS’ RESPONSE: We are very pleased to hear that the Reviewer has found our data analysis to be comprehensive and the manuscript to be well-written.
REVIEWER’S COMMENT: The concept of sleep satisfaction is certainly important, especially given the emergence of "orthosomia" as mentioned by the authors. However, objective sleep quality measures are also necessary and sufficient attributes of sleep satisfaction.
AUTHORS’ RESPONSE: We agree with the Reviewer that that objective sleep quality measures (i.e., self-reported proxy-quantifiable indices) are also necessary measures, and we indicate this in page 21 “…we surmise that when it comes to measuring sleep satisfaction among general, non-clinical populations, a shift away self-reported proxy-quantifiable indices, towards global perceived measures of sleep is likely apt. On the other hand, physiological and even proxy-quantifiable measures of sleep have relevance for populations experiencing sleep disorders or when specific dimensions of sleep need to be assessed”.
REVIEWER’S COMMENT: In Table 5, the few attributes unique to "sleep satisfaction", such as sleep environment, appear vague terms. The authors could elaborate more on what is the true biological/psychological mechanism underlying the distinction between "sleep satisfaction" and "sleep quality". In other words, are those attributes truly necessary or just arise due to pure correlation?
AUTHORS’ RESPONSE: While Table 5 presents information in brief, we elaborate on each attribute in the manuscript. We elaborate on what is meant by “sleep environment” and the other sleep satisfaction in Table 4 and in section 3.3. Phase 2 of definition generation process: identifying operationalizations of sleep satisfaction.
Our conceptualization of sleep satisfaction and its distinction from sleep quality is based on a synthesis of data reporting linkages between or among variables (these linkages include associations, predictions, differences, or changes). The linkages are discussed in section 3.2. Phase 1 of definition generation process: identifying factors (correlates, antecedents, consequences) associated with sleep satisfaction and in presented in Tables 2 and 3.
REVIEWER’S COMMENT: The authors should perform a through proofreading of the manuscript. A few mistakes in grammar is noted, for example:
- Page 17 line 1: missing "in".
- Page 17 line 26: missing "to".
- Page 18: Table 5/A5: "health behaviour" should be "healthy behavior"?
AUTHORS’ RESPONSE: We thank the Reviewer for spotting grammatical mistakes. The manuscript has now been edited by the authors.
Reviewer 2 Report
Comments and Suggestions for Authors
Congratulations for the paper! The paper is really well written, interesting topic and correct way to proceed. I agree with you in choosing the rapid review.
I accept the paper in the present form.
Author Response
We thank the Reviewer for reading our manuscript and we are delighted to hear that they found value in our research.